# Essential Roles of *Exocyst Complex Component 3-like 2* on Cardiovascular Development in Mice

**DOI:** 10.3390/life12111730

**Published:** 2022-10-28

**Authors:** Chisato Watanabe, Hirotoshi Shibuya, Yusuke Ichiyama, Eiichi Okamura, Setsuko Tsukiyama-Fujii, Tomoyuki Tsukiyama, Shoma Matsumoto, Jun Matsushita, Takuya Azami, Yoshiaki Kubota, Masahito Ohji, Fumihiro Sugiyama, Satoru Takahashi, Seiya Mizuno, Masaru Tamura, Ken-ichi Mizutani, Masatsugu Ema

**Affiliations:** 1Department of Stem Cells and Human Disease Models, Research Center for Animal Life Science, Shiga University of Medical Science, Seta Tsukinowa-cho, Otsu 520-2192, Japan; 2Technology and Development Team for Mouse Phenotype Analysis, RIKEN BioResource Research Center, Tsukuba 305-0074, Japan; 3Department of Ophthalmology, Shiga University of Medical Science, Seta Tsukinowa-cho, Otsu 520-2192, Japan; 4Department of Anatomy, Keio University School of Medicine, Tokyo 160-8582, Japan; 5Laboratory Animal Resource Center in Transborder Medical Research Center, University of Tsukuba, Tsukuba 305-8575, Japan; 6Department of Anatomy and Embryology, Faculty of Medicine, University of Tsukuba, Tsukuba 305-8575, Japan; 7International Institute for Integrative Sleep Medicine (WPI-IIIS), University of Tsukuba, Tsukuba 305-8575, Japan; 8Laboratory of Stem Cell Biology, Graduate School of Pharmaceutical Sciences, Kobe Gakuin University, Kobe 650-8586, Japan

**Keywords:** exocyst complex, exocyst complex component 3, cardiovascular development, hemorrhage, knock-out mouse, *Exoc3l2*

## Abstract

Angiogenesis is a process to generate new blood vessels from pre-existing vessels and to maintain vessels, and plays critical roles in normal development and disease. However, the molecular mechanisms underlying angiogenesis are not fully understood. This study examined the roles of *exocyst complex component (Exoc) 3-like 2 (Exoc3l2)* during development in mice. We found that *Exoc3l1, Exoc3l2, Exoc3l3* and *Exoc3l4* are expressed abundantly in endothelial cells at embryonic day 8.5. The generation of *Exoc3l2* knock-out (KO) mice showed that disruption of *Exoc3l2* resulted in lethal *in utero*. Substantial numbers of *Exoc3l2* KO embryos exhibited hemorrhaging. Deletion of *Exoc3l2* using Tie2-Cre transgenic mice demonstrated that *Exoc3l2* in hematopoietic and endothelial lineages was responsible for the phenotype. Taken together, these findings reveal that *Exoc3l2* is essential for cardiovascular and brain development in mice.

## 1. Introduction

Angiogenesis is a dynamic process to generate new blood vessels via sprouting from pre-existing vessels, followed by the remodeling of generated lumenized endothelial tubes to mature into hierarchical networks. Subsequently, pericytes or vascular smooth muscle cells cover the lumenized endothelial tubes and provide stability. Angiogenesis plays critical roles in normal development and diseases such as cancer [1,2]. These processes are controlled by many genes, such as *Vegf*/*Vegfr*, *Angiopoietin*/*Tie2, PDGF-B/PDGFR*, *TGF-β**/ALK*, *Notch*/*Dll, Wnt/Frizzled* and other signaling molecules [1,2].

The exocyst complex comprises eight subunits, including Exoc 1–8 [3,4,5]. Members of the exocyst complex were first found as genes implicated in polarized exocytosis in *Saccharomyces cerevisiae* during the genetic screening of secretory mutants [6]. The Exoc1 to Exoc8 subunits are conserved among mammalian species. Notably, *Exoc3* has four homologous genes, called *Exoc3-like 1 (Exoc3l1), Exoc3-like 2 (Exoc3l2), Exoc2-like 3 (Exoc3l3*, also called *tumor necrosis factor, alpha-induced protein 2*, *Tnfaip2*) and *Exoc3-like 4 (Exoc3l4)* in vertebrates. *Exoc3l2* is associated with human developmental anomalies [7,8,9], but its precise developmental functions remain unclear.

In previous work utilizing the *fetal liver kinase 1* knock-out (*Flk1* KO) fetus lacking endothelial cells, we showed that a class of genes were abundantly expressed in developing endothelial cells in mice. Some of the orthologous human genes, including *EXOC3L1*, have been shown to function during angiogenesis in human umbilical vein endothelial cells, although most of them have yet to be functionally characterized [10].

Here, we report the generation of KO animals for *Exoc3l2*, and initial phenotypic characterization. Most of the *Exoc3l2* KO embryos died in utero and showed hemorrhage, abnormal heart and brain development. Conditional KO animals lacking *Exoc3l2* in hematopoietic and endothelial lineages showed similar phenotypes, such as hemorrhage and heart defects, indicating that *Exoc3l2* in hematopoietic and endothelial lineages is responsible for normal cardiovascular development. Inducible KO in endothelial cells during postnatal retinal development resulted in normal angiogenesis in the retina, indicating that *Exoc3l2* is dispensable for postnatal angiogenesis in the retina. Taken together, these results indicate that *Exoc3l2* is essential for normal cardiovascular development in mice. 

## 2. Materials and Methods

### 2.1. Animals

Wild-type ICR (CD1) mice were purchased from Japan SLC Inc. (Hamamatsu, Shizuoka, Japan). Genetically modified mice targeting *Exoc3l2* were created by the animal facility in the University of Tsukuba. In brief, 5 ng/ul *pX330-mC* plasmid DNA carrying Cas9-Cdt1 fusion and respective gRNA for *Exoc3l2* together with 10 ng/ul donor DNA were injected into fertilized eggs from C57BL/6J mice according to a previous report [11]. Mice were transferred to Shiga University of Medical Science. *Exoc3l2* KO mice were initially maintained in the C57BL/6J background and then outbred with the ICR background for further phenotypic analysis. Genotyping PCR analysis for *Exoc3l2* mice was performed using a Gotaq G2 Green Master mix (Cat # M7822; Promega, Madison, WI, USA) with the primers listed in Appendix A. 

### 2.2. Reverse Transcription (RT)-Quantitative (q) PCR 

Total RNA was extracted using a RNeasy Micro kit (Cat # 74004; Qiagen, Hilden, Germany), followed by cDNA synthesis using the ReverTra Ace kit (Cat # TRT-101X5; Toyobo, Osaka, Japan) according to the manufacturer’s instructions. Real-Time PCR was performed with the Thermal Cycler Dice Real Time System (TaKaRa Bio Inc., Otsu, Shiga, Japan) and Thunderbird SYBR qPCR Mix (Cat # QPS-201T; Toyobo). Data were normalized against the expression of *Hprt*. Primer sequences are listed in Appendix A.

### 2.3. Single Cell (sc) RNA-Seq Analysis

scRNA-seq data for E8.5 mouse embryo (GSE186069) were downloaded from GEO database [12]. We performed conventional scRNA-seq processing using Seurat_v4.1.1 by R (https://www.r-project.org/, accessed on 10 October 2022). At first, we extracted “A2-” data set from whole scRNA-seq data for the analysis for *Exoc3l2*. Extracted scRNA-seq data were filtered to remove the dead cell population and empty droplets using cell annotation information described in C. Qiu et al. After the normalization of UMI counts by the total counts per cells, chose 2000 most highly variable genes for scaling the expression, then PCA was performed. To find the significant PCs, efficacious PC threshold was calculated by (1) cumulative %stdev of PCs > 90 and %variation associated with PCs < 5, and (2) last point of change of %PC variation > 0.1. We selected minimum value between (1) and (2) as the significant PC for this analysis. Finishing pre-processing of clustering, k-NN graph was generated by Louvain clustering using the top 14 of PCs (resolution = 0.4) and UMAP was drawn. Endothelial marker genes (*Pecam1* and *Cdh5*) and *Exoc3l2* expressions were shown on the same UMAP. 

### 2.4. Immunohistochemistry

Mice were sacrificed by cervical dislocation. Embryos were then dissected and fixed in 4% paraformaldehyde for 2–4 h at 4 °C, as described previously [13] For whole-mount immunohistochemistry, after washing twice in PBS, embryos were permeabilized in 0.5% TritonX-100 (CAS: 9002-93-1; Sigma-Aldrich, St. Louis, MO, USA) in Phosphate-Buffered Saline (PBS) for 30 min at 4 °C. Permeabilized embryos were then blocked in blocking solution containing 10% donkey serum (Cat # IHR-8135, Immuno Bioscience, Mukilteo, WA, USA), 0.2% bovine serum albumin (A7906-50G; Sigma-Aldrich, St. Louis, MO, USA) and 0.01% Tween20 in PBS (0.01% PBT)(Code: 28353-85; Nacalai Tesque, Inc., Kyoto, Japan) for 1 h at 4 °C, followed by incubation overnight at 4 °C. The embryos were further treated with rat anti-Pecam1 antibody (1:100; Cat # 553370; BD Pharmingen, Franklin Lakes, NJ, USA) and anti-GFP rabbit polyclonal antibody Alexa Fluor 488 conjugate (1:200; Cat # A21311; Life Technologies, Inc., Carlsbad, CA, USA). After three washes with 0.2% PBT, embryos were incubated with Cy3-conjugated donkey anti-Rat IgG (H + L) secondary antibody (Cat # 712-165-150; Jackson Immunoresearch, West Grove, PA, USA) for 1–3 h at room temperature. Embryos were then washed three times with 0.2% PBT. Nuclei were stained with 10 µg/mL Hoechst33342 (Cat # H3570; Molecular Probes Inc., Eugene, OR, USA) for 20 min at 4 °C. 

Brain sections were immunostained as described previously [14]. In brief, 200 µm thick coronal sections were prepared from telencephalon. Brains were fixed in 1% paraformaldehyde overnight at 4 °C, embedded in agarose gel, and cut using a vibrating microtome (Leica VT1200). Vibratome sections were treated with blocking buffer (10% donkey serum and 0.3% Triton X-100) overnight at 4 °C, followed by incubation with primary antibodies [anti-Pecam1 (# MAB1398Z; Millipore, Burlington, MA, USA), anti-platelet-derived growth factor-beta receptor (PDGFRβ) (# 14-1402-82; eBioscience), anti-phospho-histone H3 (# 9701; Cell Signaling)] diluted in the same buffer overnight at 4 °C. Immunolabeled sections were incubated with secondary antibodies overnight at 4 °C. After washing, sections were mounted under a cover glass with mounting medium (Fluoro-Keeper Antifade Reagent, Nacalai). Images were acquired on a confocal microscope (FV3000, Olympus) and processed using Adobe Photoshop.

For cryosections, embryos were mounted in OCT (Surgipath FSC 22 Blue Frozen section compound; Leica, Wetzlar, Germany). Tissue blocks were sectioned using a cryostat (CM1860 UV; Leica) and then immunostained with rabbit anti-mouse Type-IV collagen (1:500; Cat # 2150-1470; AbD Serotec, Hercules, CA, USA), rabbit anti-mouse Lyve1 antibody (Cat# ab14917, abcam, Cambridge, UK) and rat anti-mouse Ter119 antibody (Cat # 14-5921-82; Thermo Fisher Scientific, Waltham, MA, USA) as previously described [15]. 

### 2.5. Immunostaining of Whole-Mount Retinas 

VE-cad-CreER mice [16] were mated with floxed *Exoc3l2* mice to obtain conditional *Exoc3l2* KO mice. Forty μg 4-hydroxytamoxifen was subcutaneously injected at postnatal day 2, 3.5 and 5. Littermates were used as controls.

Whole-mounted retinas were prepared according to a previous report [17]. Briefly, enucleated eyes were fixed for 20 min in 4% paraformaldehyde at room temperature. A small hole was made in the cornea using a 27-gauge needle, and a circular incision was made using fine scissors. Then, retinal cups were dissected from the eyes and postfixed for 30 min in 4% paraformaldehyde at 4 °C. The primary antibodies used were rabbit anti-GFP (Alexa 488-conjugated; Molecular Probes), biotinylated isolectin B4 (1:500; B-1205; Vector Laboratories), mouse anti-Ter119 (1:250; MAB1125; R&D Systems, Inc., Minneapolis, MN, USA), rabbit anti-neuron-glial antigen-2 (NG2) (1:200; AB5320; Millipore-Sigma, Burlington, MA, USA) and hamster anti-Pecam1 (1:1000; ab119341; Abcam, Cambridge, UK). The secondary antibodies were suitable species-specific secondary antibodies (Jackson ImmunoResearch Laboratories, Inc., West Grove, PA, USA) or streptavidin coupled to Alexa Fluor dyes (Invitrogen, Waltham, MA, USA).

### 2.6. Image Acquisition 

Confocal images were acquired on a Leica TCS-SP8 (Leica). Embryos were mounted in PBS on glass-bottom dishes (Cat # 3971-035; IWAKI, Tokyo, Japan). Fluorescence was excited with a 405 nm UV laser to detect Hoechst 33342, a 552 nm laser to detect Cy3, and a 488 nm laser to detect Alexa Fluor 488. The hybrid detector HyD (Leica) was used for signal amplification. Projection images were created by 3D viewer software (Leica) from z-stack images every 2–4 µm.

Whole retinas were mounted using ProLong™ Gold Antifade Mountant (Thermo Fisher Scientific). Fluorescence was excited with a 647 nm laser for Cy5, a 552 nm laser for Cy3, a 488 nm laser for Alexa Fluor 488, and a 405 nm UV laser for Hoechst 33342 and DAPI. Z-stack images were obtained every 5 µm.

### 2.7. Flowcytometry Analysis

Embryos were incubated with 0.3% collagenase (Cat # 038-22361; Wako, Osaka, Japan) at 37 °C for 15 min. After washing, single cells were exposed to allophycocyanin-conjugated anti-mouse Pecam1 antibody (MEC13.3, Cat #102510; BioLegend, San Diego, CA, USA). Cells were analyzed by FACS calibur (Becton Dickinson, San Jose, CA, USA) and the data were processed by the FlowJo software (TreeStar, Ashland, OR, USA).

### 2.8. CT imaging Analyses

Mouse embryos were fixed with Bouin’s fixative solution, a 15:5:1 mixture of water-saturated picric acid, concentrated formalin and concentrated acetic acid. Samples were washed with 70% EtOH and soaked in 1% phosphotungstic acid in 70% EtOH as contrast agent. Then, samples were scanned with Scanxmate-E090S (Comscantechno, Yokohama, Kanagawa, Japan), and rotated 360° in steps of 0.3°, generating 1200 projection images of 992 × 992 pixels. Reconstruction of the CT images was carried out using the filtered back projection. The cardiovascular system and soft tissues were visualized using maximum intensity projection and multi-planar reconstruction, respectively, in OsiriX software (www.osirix-viewer.com, accessed on 5 June 2021).

### 2.9. Statistical Analysis

The differences in Mendelian ratio were analyzed by the chi-square test, with *p* < 0.05 considered significant.

Student’s *t*-test was used for statistical analysis. Data were presented by means and standard errors. Statistical significance was determined at *p* < 0.05.

## 3. Results

### 3.1. Expression of Exoc3l Members and Generation of Exoc3l2 KO Mice

Our previous work showed that several *exocyst complex component 3-like* members were abundantly expressed in developing endothelial cells [10], but it is not known whether they play any physiological role during development. To address this issue, the gene expression patterns of *Exoc3l1*, *Exoc3l2*, *Exoc3l3* and *Exoc3l4* were determined by RT-qPCR analysis using cDNA from a wild-type (WT) fetus at embryonic day 8.5 (E8.5), *Flk1* KO fetus at E8.5 (which lacks endothelial cells), endothelial cells purified from a *Flk1^+/GFP^* knock-in fetus at E8.5, and peripheral blood cells isolated at E9.5 (Figure 1a). RT-qPCR analysis showed that *Exoc3l1*, *Exoc3l2*, *Exoc3l3* and *Exoc3l4* were abundantly expressed in WT but *Flk1* KO fetuses at E8.5 were not. They were also expressed more abundantly in endothelial cells sorted from a WT fetus at E8.5, but not in peripheral blood cells at E9.5 (Figure 1a). These results indicated that *Exoc3l1*, *Exoc3l2, Exoc3l3* and *Exoc3l4* are expressed abundantly in endothelial cells at E8.5, in a sharp contrast with the ubiquitous pattern of other *exocyst complex* members, such as *Exoc3* (Figure 1a). Analysis with publicly available scRNA-seq data on E8.5 embryo also confirmed the endothelial expression of *Exoc3l2* (Appendix A).

To investigate whether *Exoc3l2* plays roles in developing endothelial cells, KO mice for *Exoc3l2* were created by deleting most of the protein coding exons (Figure 1b). For the *GFP* knock-in allele, a linker sequence, GSG, a self-cleaving peptide sequence, P2A, and *GFP* were introduced into exon2 (Figure 1b). 

Whole-mount immunohistochemistry of an *Exoc3l2^+/GFP^* embryo at E9.5 showed GFP expression considerably overlapped with that of Pecam1, an endothelial marker, and GFP expression was stronger in microcapillary rather than in moderately sized blood vessels (Figure 1c). GFP expression was also observed in non-endothelial cells, which are most likely cranial neural crest cells (Figure 1c). Since GFP expression is low or negative in large blood vessels, GFP expression was evaluated in Pecam1-positive endothelial cells by FACS. This analysis found that 15% of Pecam1-positive cells expressed GFP (Figure 1d), suggesting that subpopulation of Pecam1-positive endothelial cells express *Exoc3l2*, although the biological roles of the subpopulation remain unclear. 

### 3.2. Phenotypes of Exoc3l2 KO Embryo

When *Exoc3l2* heterozygotes were crossed with each other, *Exoc3l2* KO pups were never born, suggesting an important role for *Exoc3l2* during embryonic development (Figure 2a). *Exoc3l2* KO embryos were present at a Mendelian ratio until E15.5 (*p*(X^2^) = 0.9131 by the chi-square test), although a substantial number (9 out of 17) were dead (Figure 2a). *Exoc3l2* KO but not WT embryos exhibited extensive hemorrhages at E15.5 (9 out of 17) (Figure 2a,b). Histological analysis with hematoxylin and eosin indicated that WT embryos had most, if not all, peripheral red blood cells inside blood vessels, and many red blood cells were seen beneath the skin in *Exoc3l2* KO embryo (Figure 2b). Immunohistochemistry to highlight hemorrhaged red blood cells showed that Ter119-positive peripheral red blood cells were distributed outside blood vessels as well as inside, while most red blood cells were located inside in WT embryo (Figure 2c). Micro-CT analysis was used to inspect the location of hemorrhaging throughout body. The µ-CT contrast agent (CECT) is a powerful tool for high-throughput, high-resolution embryonic phenotyping, and is used for embryonic lethal phenotyping at the International Mouse Phenotyping Consortium [18]. In addition, CTEC can visualize the cardiovascular system via the possible detection of blood components such as clots [19]. The CECT indicated that hemorrhaging occurred in internal organs, including the liver as well as under the skin (Figure 2d). 

### 3.3. Phenotypes of Tie2-Cre: Exoc3l2 Conditional Knock-Out Embryos

Although conventional KO of *Exoc3l2* in mice showed that the lack of *Exoc3l2* resulted in embryonic lethality, possibly due to hemorrhaging, *Exoc3l2* is expressed in non-endothelial cells in the head region, likely to be cranial neural crest cells, and potentially non-endothelial cells (Figure 1c). To investigate whether endothelial *Exoc3l2* is critical for proper embryonic development, a floxed allele for *Exoc3l2* was generated by introducing loxP sequences at both ends of exon 3 (Figure 3a). Exon3 was floxed, because deletion of exon3 leads to a frameshift and production of a truncated protein (209 amino acids) lacking entire Sec6 domain which is conserved among Exoc genes. Crossing *Exoc3l2^flox/flox^* mice with *Tie2-Cre*: *Exoc3l2^+/^**^Δ^* mice (expressing Cre in hematopoietic and endothelial lineages) generated *Tie2-Cre*:: *Exoc3l2^flox/^**^Δ^* embryos (hereafter denoted *Exoc3l2**^ΔHEC^*), which lack *Exoc3l2* in hematopoietic and endothelial lineages (Figure 3a,b). Gross observation indicated that a significant number of *Exoc3l2**^ΔHEC^* embryos showed embryonic lethality around E17.5 and exhibited hemorrhaging under the skin, similar to the phenotype of *Exoc3l2* KO embryos (Figure 3b,d). Previous work reported that blood cells could be observed inside lymphatic vessels because of improper fusion of the vascular system and lymphatic vessels [20,21]. However, this scenario is not the case in the current study because there were no blood cells inside lymphatic vessels in *Exoc3l2**^ΔHEC^* embryos (Figure 3e). 

CECT analysis was used to inspect the hemorrhage phenotype more globally in deeper tissues of the embryos. Although WT embryos showed no detectable hemorrhaging (Figure 4a–c), *Exoc3l2**^ΔHEC^* embryo showed hemorrhaging from various blood vessels throughout the body, including large vessels (Figure 4d–h). These data suggest that *Exoc3l2* function in hematopoietic and endothelial is essential for vessel development and/or homeostasis. 

### 3.4. Postnatal Roles of Exoc3l2 in Retinal Vascular Development

Since *Exoc3l2* plays a critical role in vascular stability during embryonic development, we predicted that *Exoc3l2* may participate in vascular stability during the postnatal stage. To investigate a role of *Exoc3l2* on angiogenesis during postnatal development, we have used an angiogenesis model in retina, because retinal angiogenesis starts from the birth and often used to measure the angiogenic ability of endothelial cells. *Exoc3l2-GFP* was weakly expressed in retinal endothelial cells (Figure 5a). Therefore, *VE-cadherin-CreER*: *Exoc3l2^+/GFP^* mice and *Exoc3l2 ^flox/flox^* mice were crossed to conditionally delete *Exoc3l2* in the postnatal stage by treatment with tamoxifen (Figure 5b). This strategy generated *VE-cadherin-CreER::Exoc3l2* conditional KO mice (hereafter denoted *Exoc3l2^i^**^ΔEC^*), which lack *Exoc3l2* in the postnatal endothelial lineage (Figure 5b). Immunohistochemistry for Pecam1 and NG2, a pericyte marker in the retina showed that radial growth, pericyte coverage and filopodia number were similar in WT and *Exoc3l2^i^**^ΔEC^* mice, indicating that *Exoc3l2* is dispensable for postnatal angiogenesis in retina (Figure 5c).

### 3.5. Roles of Exoc3l2 in Heart and Brain Development

Since *Exoc3l2* is associated with Dandy–Walker malformation in Meckel–Gruber syndrome in human [7,8,9], *Exoc3l2* may play other physiological roles in other cell types than endothelial cells. To address this issue, CECT was used to examine the phenotype of *Exoc3l2**^ΔHEC^* KO to determine potential *Exoc3l2* functions during embryogenesis. This analysis used three and six *Exoc3l2**^ΔHEC^* KO embryos at E12.5 and E15.5, respectively (Figure 6). Some *Exoc3l2^ΔHEC^* KO embryos showed ventricular septal defects (2 out of 3 examined) (Figure 6b, arrows) and hypoplasia of myocardium (2 out of 3 examined) (Figure 6b,c, arrowheads) in the heart at E12.5. Furthermore, *Exoc3l2**^ΔHEC^* KO embryos had morphological brain anomalies in the telencephalon and cerebellum at E15.5 (arrowheads in Figure 6e,f). These results indicated that *Exoc3l2* in hematopoietic and endothelial lineages play a role in normal heart and brain development.

Our previous report indicated endothelial cells regulate the proliferation and differentiation of neural stems [14]. Therefore, the numbers of apical and basal progenitors (phospho-histone H3-positive neural stem and progenitors) were quantified in the E16.5 neocortex (Figure 7). There were no significant differences between apical and basal progenitor numbers in WT and *Exoc3l2**^ΔHEC^* KO embryos, suggesting that endothelial *Exoc3l2* regulates brain development through other pathways.

## 4. Discussion

In this paper, we report for the first time that *Exoc3l2* is required for normal angiogenesis, heart and brain development, and the lack of *Exoc3l2* resulted in hemorrhages, abnormal heart and brain development. By using conditional knock-out approaches, we demonstrated that *Exoc3l2* function in hematopoietic and endothelial lineages is indispensable for normal angiogenesis, heart and brain development. In contrast with essential roles of *Exoc3l2* during embryonic development, *Exoc3l2* is not required for postnatal retinal angiogenesis. However, role of *Exoc3l2* on pathological angiogenesis such as tumour angiogenesis remains unclear.

Since *Exoc3l2* is deleted in both hematopoietic and endothelial lineages of *Exoc3l2**^ΔHEC^* embryos, it is not clear which lineage is responsible for the hemorrhages and abnormal heart development. Further analysis with specific Cre-expressing mice for hematopoietic or endothelial cells will be required for the clarification.

Currently, the molecular and cellular mechanism of how *Exoc3l2* regulates angiogenesis, heart and brain development is not known. Previous reports show that proper interaction between vascular endothelial cells is needed for normal angiogenesis, and disrupted interaction results in hemorrhage [22]. Normal pericyte recruitment on endothelial cells mediated by smooth muscle cells or pericytes is also required for vascular stabilization [1,2]. Exocyst complex members regulate E-cadherin trafficking from recycling endosomes to plasma membranes, E-cadherin clustering [23], epithelial polarization [24], Notch signaling and primary ciliogenesis [25]. Therefore, *Exoc3l2* may function as a regulator for cadherin trafficking and Notch signaling in endothelial cells, to control vascular stabilization. Since Exoc genes are implicated in vesicle trafficking, *Exoc3l2* may affect various organ development other than cardiovascular and brain development via paracrine manner.

Qu and coworkers showed that endocardial Tie2 regulates myocardial trabeculation through the retinoic acid pathway [26]. Therefore, heart development may be regulated by Exoc3l2 through Tie2. Barkefors and coworkers reported that Exoc3l2 is expressed in vascular endothelial cells and forms a complex with Exoc4, a component of exocyst, and Exoc3l2 knock-down resulted in reduced phosphorylation of Flk1 and decreased sprouting of endothelial cells in vitro [27]. It will be interesting to investigate whether the VEGF-Flk1 pathway is affected in *Exoc3l2* KO endothelial cells in future research.

Shaheen and coworkers reported that a patient with a biallelic *EXOC3L2* (human ortholog of murine *Exoc3l2*) mutation had renal and neurological phenotypes called the Dandy–Walker malformation in Meckel–Gruber syndrome [7]. There are two other reports indicating that biallelic mutations of *EXOC3L2* causes developmental anomalies, such as peritrigonal and cerebellar abnormalities, with enlargement of ventricular trigones, pituitary hypoplasia, severe renal dysplasia and bone marrow failure [7]. Since we have shown that conditional *Exoc3l2* KO embryos exhibited developmental anomalies in the blood vessels, brain and heart, *Exoc3l2* KO mice provide a valuable animal model for the Dandy–Walker malformation in Meckel–Gruber syndrome, although further analysis on other organs and tissues of *Exoc3l2* mutant mice will be needed.

Recent genome-wide association studies revealed an association between the *EXOC3L2* rs597668 variant and Alzheimer’s disease in European and Asian populations [28,29,30,31]. Since *Exoc3l2* was shown to contribute to brain development by our study, it will be interesting to examine whether *Exoc3l2* also plays a role in human brain disorders, such Alzheimer’s disease.

Taken together, we have shown that *Exoc3l2* is essential for cardiovascular and brain development. It will be interesting to investigate the molecular mechanism by *Exoc3l2* and apply the knowledge to therapy for human diseases.

## Figures and Tables

**Figure 1 life-12-01730-f001:**
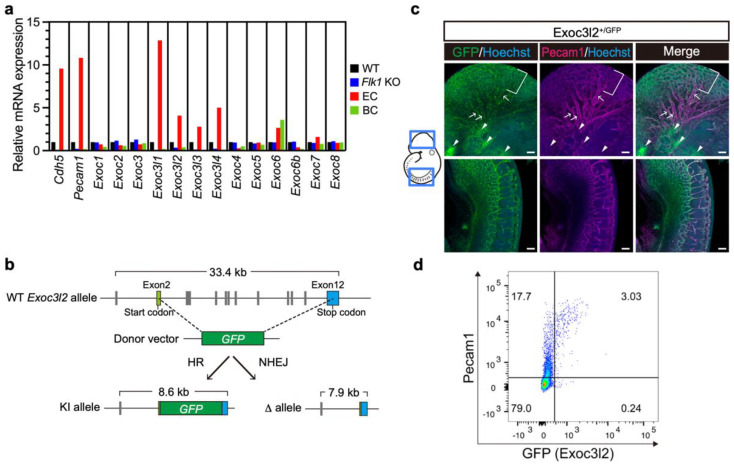
Expression of *exocyst complex component 3-like member 2 (Exoc3l2*) in developing endothelial cells. (**a**) RT-qPCR analysis of *exocyst complex* genes. WT, wild-type fetus at embryonic day 8.5 (E8.5); KO, *fetal liver kinase 1* knock-out fetus at E8.5; EC, endothelial cells purified from E8.5 whole embryos; BC, peripheral blood cells from E9.5. (**b**) Schematic representation of mutant alleles for *Exoc3l2* in mouse. HR; homologous recombination, NHEJ; Non-homologous end-joining, KI; knock-in. (**c**) Expression of GFP and Pecam1 in *Exoc3l2^+/GFP^* endothelial cells at E9.5. Brackets and arrows in upper panel show strong GFP signal in microcapillaries and weak GFP signal in blood vessels, respectively. Note that GFP expression is weak or absent in larger blood vessels. Arrowheads indicate GFP signal, which is Pecam1 negative. Scale bar: 100 µm. (**d**) Flowcytometry analysis of *Exoc3l2^+/GFP^* embryos at E9.5 with an anti- Pecam1 antibody.

**Figure 2 life-12-01730-f002:**
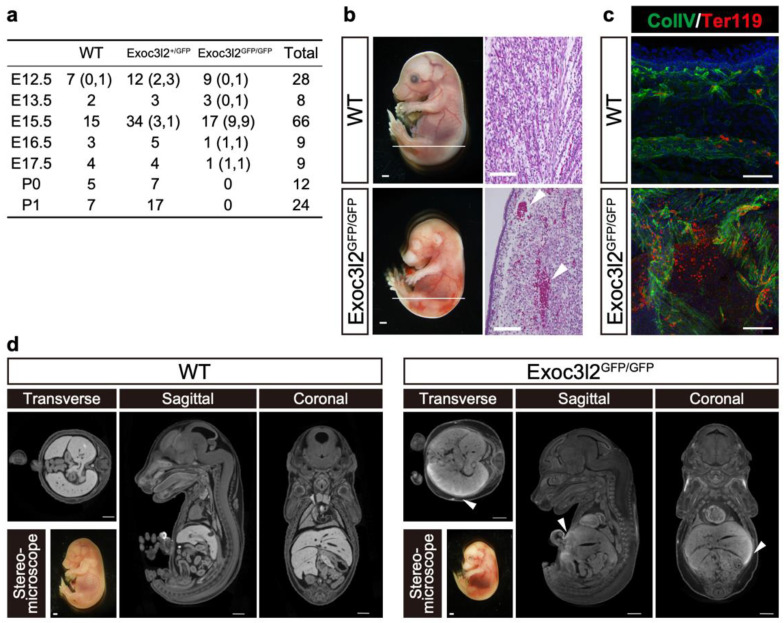
Hemorrhaging in *exocyst complex component 3-like member 2*-knock-out (*Exoc3l ^GFP/^**^GFP^*) embryos. (**a**) Summary of genotyping embryos/pups from crosses between *Exoc3l2^+/^**^GFP^* mice. Numbers in left and right side in brackets indicate dead and hemorrhaging embryos, respectively. Numbers without bracket indicates alive embryos without hemorrhaging. (**b**) Lateral view of wild-type (WT) and *Exoc3l2^GFP/^**^GFP^* embryos at embryonic day 15.5 (E15.5) and histology. Arrowheads indicate hemorrhages under skin. Scale bars in the lateral view of embryo and hematoxylin and eosin-stained sections indicate 1 mm and 100 µm, respectively. (**c**) Immunohistochemistry of WT and *Exoc3l2^GFP/^**^GFP^* embryos with anti-ColIV (collagen IV, endothelial marker) and anti-Ter119 (red blood cell marker) antibodies at E15.5. (**d**) Micro-CT analysis of WT and *Exoc3l2^GFP/^**^GFP^* embryo samples. Arrowheads indicate hemorrhages. Scale bars: 1 mm.

**Figure 3 life-12-01730-f003:**
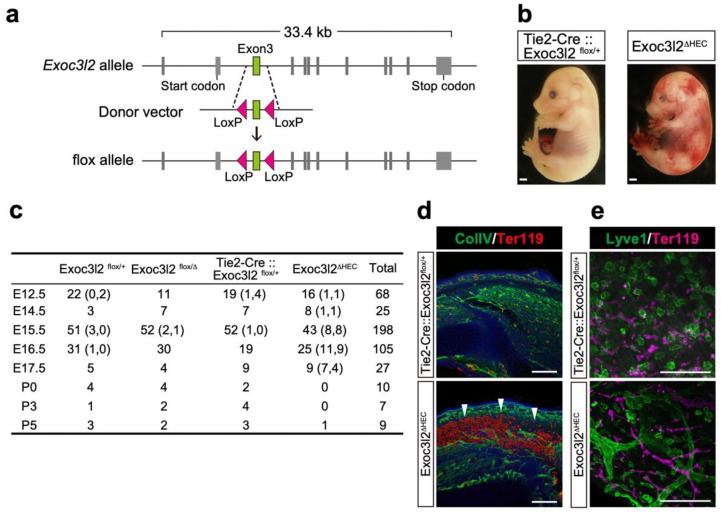
Hemorrhaging in embryos lacking *exocyst complex component 3-like member 2* in hematopoietic and endothelial lineages (denoted *Exoc3l2**^ΔHEC^*): (**a**) Schematic representation of a conditional *Exoc3l2* allele. (**b**) Lateral views of wild-type (WT) and *Exoc3l2**^ΔHEC^* embryos at embryonic day 15.5 (E15.5). Scale bars: 1 mm. (**c**) Genotypes of littermates at respective developmental stages derived from crossing *Tie2-Cre::Exoc3l2^+/^**^Δ^* (*Exoc3l2* heterozygotes (*Exoc3l2**^+/Δ^*) expressing Cre under Tie2 promoter) and *Exoc3l2^flox/flox^* mice. Numbers in left and right side in brackets indicate dead and hemorrhaging embryos, respectively. Numbers without bracket indicates alive embryos without hemorrhaging. (**d**) Immunohistochemistry of WT and *Exoc3l2**^ΔHEC^* embryos at E15.5 with anti-CollV (collagen IV) and -Ter119 antibodies. White arrowheads indicate hemorrhaging. (**e**) Immunohistochemistry of WT and *Exoc3l2**^ΔHEC^* embryos at E15.5 with anti-Lyve1 (lymphatic endothelial cell marker) and -Ter119 antibodies.

**Figure 4 life-12-01730-f004:**
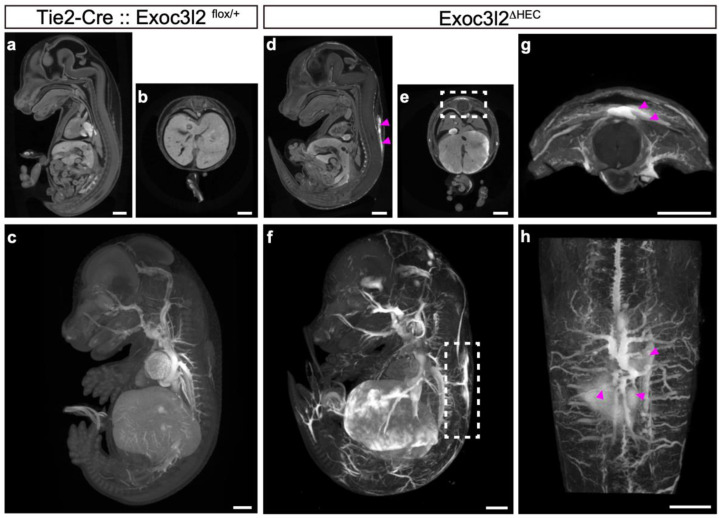
CT imaging analyses of the cardiovascular system in mouse embryos lacking *exocyst complex component 3-like member 2* in hematopoietic and endothelial lineages (*Exoc3l2^ΔHEC^*). The multi-planar reconstruction of µ-CT images at embryonic day 15.5 (E15.5) of wild-type (**a**–**c**) and *Exoc3l2^ΔHEC^* (**d**–**f**) mice are shown. The dotted square in (**e**) indicates the corresponding region of Figure (**g**). *Exoc3l2^ΔHEC^* mice exhibited the hemorrhage phenotype (arrowheads in (**d**,**g**,**h**)). The maximum intensity projection of µ-CT images of wild-type (**c**) and *Exoc3l2^ΔHEC^* (**f**,**h**) mice at E15.5 are shown. Panel (**h**) shows the dorsal view image of the dotted square in **f**. Hemorrhages from large vessels were observed in *Exoc3l2^ΔHEC^* mice (arrowheads in (**h**)).

**Figure 5 life-12-01730-f005:**
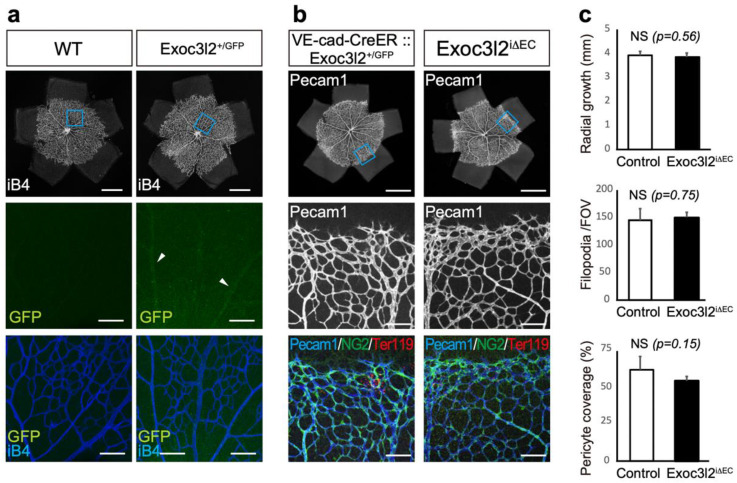
Retinal angiogenesis in neonates with conditional disruption of *exocyst complex component 3-like member 2* in the postnatal endothelial lineage (denoted *Exoc3l2^i^**^ΔEC^*). (**a**) Retinal expression of GFP in *Exoc3l2^+/GFP^* mice at postnatal day 7 (P7). GFP was present in the retinal vasculature as indicated by arrowheads. Scale bars in whole retina and magnified images indicate 1 mm and 100 µm, respectively. (**b**) Evaluation of retinal angiogenesis of *Exoc3l2^i^**^ΔEC^* mice at P6. Scale bars in whole retina and magnified images indicate 1 mm and 100 µm, respectively. Tamoxifen was administered to *VE-cadherin-CreER:Exoc3l2^GFP/flox^* mice at P2, 3.5 and 5, and pups were sacrificed at P6. (**c**) Radial outgrowth, filopodia number and pericyte coverage of control and *Exoc3l2 ^i^**^ΔEC^* neonates. NS; not significant.

**Figure 6 life-12-01730-f006:**
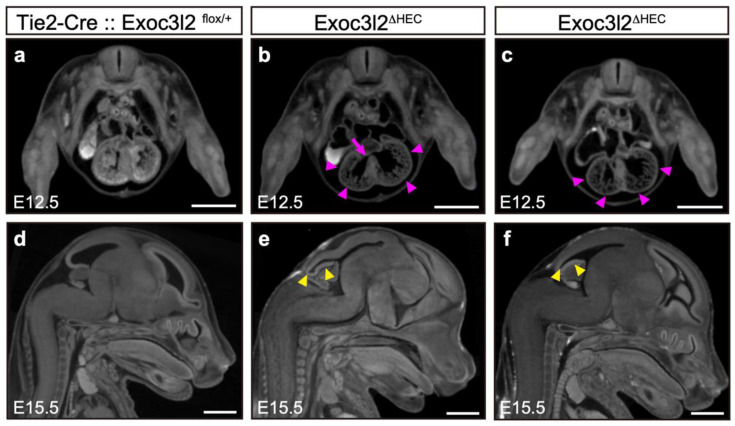
Heart and brain defects in mice lacking *exocyst complex component 3-like member 2* in hematopoietic and endothelial lineages (*Exoc3l2^ΔHEC^*). Micro-CT images of the heart at embryonic day 12.5 (E12.5) (**a**–**c**). Compared with wild-type (**a**), the hearts of *Exoc3l2^ΔHEC^* mice (**b**,**c**) showed hypoplasia of myocardium (arrowheads in purple). At this stage, some *Exoc3l2^ΔHEC^* mice exhibited ventricular septal defects (arrow). Micro-CT images of wild-type (**d**) and *Exoc3l2^ΔHEC^* (**e**,**f**) brains at E15.5. *Exoc3l2^ΔHEC^* mice exhibited abnormal brain development. Morphological anomalies in the telencephalon and cerebellum were observed in *Exoc3l2^ΔHEC^* mice (arrowheads in yellow). Scale bar: 1 mm.

**Figure 7 life-12-01730-f007:**
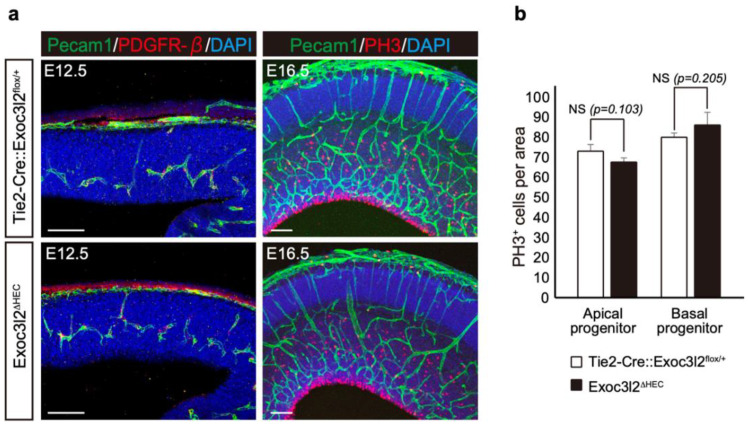
Analysis of brain development in mice lacking *exocyst complex component 3-like member 2* in hematopoietic and endothelial lineages (*Exoc3l2**^ΔHEC^*). (**a**) Immunofluorescence in the embryonic day 12.5 (E12.5) [green, Pecam1; red, platelet-derived growth factor receptor beta (PDGFR-β)] and E16.5 [green, Pecam1; red, phospho-histone H3 (PH3)] WT and *Exoc3l2**^ΔHEC^* neocortex. (**b**) Quantification of apical and basal progenitors (PH3+ neural stem and progenitors) in the E16.5 neocortex; NS, not significant; n = 5 sections from different littermates. Scale bar: 100 µm.

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
