# Peer review of "Essential Roles of Exocyst Complex Component 3-like 2 on Cardiovascular Development in Mice"

_life, 2022, doi:10.3390/life12111730_

Round 1

Reviewer 1 Report

In this manuscript, Watanabe C et al. indicated that exocyst complex component 3-like 2, Exoc 3l2, is predominantly expressed in endothelial cells in the developmental stage. Loss of exoc 3l2 caused embryonic lethality mainly due to bleeding. This reviewer did not entirely recognize the criteria of the MDPI journal series. Although this manuscript lacks any mechanistic and biochemical approach, phenotypic data presentation with global, endothelial cell-specific Tie2-care and conditional Cre were well performed and reliable. CT imaging from the embryos was excellent. This reviewer has several questions and recommendations, as shown below;

Specific comments;

1.           Fig. 1a, the author picked up the data from flk-1 knockout mice; thus, it may raise confusion. After watching the gene targeting methods from Fig. 2b, many readers realize the WT and KO in Fig. 1a simply from exoc3l2. Moreover, this reviewer felt the current explanation of EC-specific expression via Flk-1 is somewhat complicated. If the author can pick the recent publicly available single-cell RNA-seq data in mouse development, directly indicate and summarize the expression level of the Exotic family from the database. The author argues against the tie2-Cre  mediated problem; hematopoietic and endothelial cells are affected. Thus, this reviewer would like to know the expression of hemogenic angioblast on E7.5-8.5, such that the time point are missing the current Fig.1a. 

2.           Since exoc 3l2 expressed various developmental blood vessels, why the tie2-Cre mediated conditional KO indicate hemorrhage only from large vessels, as shown in Fig.4? 

3.           Fig. 5a, indicating GFP images were poor, though the rests were nicely illustrated. Thus, this reviewer could not find any specific GFP-positive signals. The author should reformat the figure. This reviewer agrees that such GFP signals were greatly diminished after the postnatal periods. Firstly, why exoc3l2 genes expression are lost postnatally? If possible, please add any possibilities to the Discussion section.

4.           Current Discussion contents are insignificant. It should rewrite completely. Overall, this author notices the possibility of poor correlations to the endothelium from the previous reports. Anything can be possible if the current author's style is kept. If such simple cases are shown throughout, this reviewer recommends verifying them via biochemical studies. Instead, this reviewer recommends profoundly thinking about the current data involving my questions from 2 and 3 and then writing in the Discussion section.

Minor comment in detail,

On page 4, line 199, 'Exoc3' is suited better than Exoc1.

Author Response

In this manuscript, Watanabe C et al. indicated that exocyst complex component 3-like 2, Exoc 3l2, is predominantly expressed in endothelial cells in the developmental stage. Loss of exoc 3l2 caused embryonic lethality mainly due to bleeding. This reviewer did not entirely recognize the criteria of the MDPI journal series. Although this manuscript lacks any mechanistic and biochemical approach, phenotypic data presentation with global, endothelial cell-specific Tie2-care and conditional Cre were well performed and reliable. CT imaging from the embryos was excellent. This reviewer has several questions and recommendations, as shown below;

Specific comments;

Fig. 1a, the author picked up the data from flk-1 knockout mice; thus, it may raise confusion. After watching the gene targeting methods from Fig. 2b, many readers realize the WT and KO in Fig. 1a simply from exoc3l2.

We thank for the comment. We have changed KO to Flk1 KO in Fig. 1a to avoid a confusion.

Moreover, this reviewer felt the current explanation of EC-specific expression via Flk-1 is somewhat complicated. If the author can pick the recent publicly available single-cell RNA-seq data in mouse development, directly indicate and summarize the expression level of the Exotic family from the database.

We thank for the comment. We have performed an analysis on scRNA-seq data which is available publicly and found that Exoc3l2 is expressed in CD31-positive endothelial cells. We included the data as Fig.S1.

The author argues against the tie2-Cre  mediated problem; hematopoietic and endothelial cells are affected. Thus, this reviewer would like to know the expression of hemogenic angioblast on E7.5-8.5, such that the time point are missing the current Fig.1a. 

We thank for the comment. It is broadly recognized that Cre in Tie2-Cre mice deletes floxed allele in both hematopoietic and endothelial cells. Please see DOI: 10.1161/ATVBAHA.118.309669

Since exoc 3l2 expressed various developmental blood vessels, why the tie2-Cre mediated conditional KO indicate hemorrhage only from large vessels, as shown in Fig.4? 

We apologize for the error. Actually, hemorrhages were seen in various blood vessels throughout body, including large blood vessels. Therefore, we have carefully re-written our manuscript to correct the text as following; various blood vessels throughout body, including large blood vessels shown in Fig. 4

Fig. 5a, indicating GFP images were poor, though the rests were nicely illustrated. Thus, this reviewer could not find any specific GFP-positive signals. The author should reformat the figure. This reviewer agrees that such GFP signals were greatly diminished after the postnatal periods. Firstly, why exoc3l2 genes expression are lost postnatally? If possible, please add any possibilities to the Discussion section.

We thank for the comment. We have added arrowheads to indicate the weak GFP signals. Regarding a role of Exoc3l2 during postnatal retina development, we presume that Exoc3l2 is not required for normal angiogenesis during post-natal stage, but for pathological angiogensis such as tumour angiogenesis. We have added the sentence in discussion section.

Current Discussion contents are insignificant. It should rewrite completely. Overall, this author notices the possibility of poor correlations to the endothelium from the previous reports. Anything can be possible if the current author's style is kept. If such simple cases are shown throughout, this reviewer recommends verifying them via biochemical studies. Instead, this reviewer recommends profoundly thinking about the current data involving my questions from 2 and 3 and then writing in the Discussion section.

We thank for the comment. According to your question 2 and 3, we have modified discussion section.

Minor comment in detail,

On page 4, line 199, 'Exoc3' is suited better than Exoc1.

We thank for the comment. We have corrected accordingly.

Reviewer 2 Report

The manuscript presents the results of a study aimed at elucidating the role of Exocyst complex component 3-like 2 on cardiovascular development in mice. The authors used state-of-the-art technology and created knockout mouse models that can serve as experimental models for the Dandy-Walker malformation in Meckel-Gruber syndrome. The manuscript deserves publication, but requires significant revision.

The present version of the title, conclusions, and overall text implies that the effect of the Exoc3l2 deletion extends only to the cardiovascular system and nothing else, but this cannot be the case based on the demonstrated functions of Exoc3l2. The content of the manuscript needs to be reviewed and the emphasis in the presentation of the results, conclusions, and title of the manuscript should be changed so that the reader can understand that the authors consider the cardiovascular system as an example of one of many tissues for demonstrating the function of the Exoc3l2 gene during the development of the organism.

Abstract: The manuscript presents results concerning not only the cardiovascular system, but also the development of brain structures. Why is not a word written in the abstract about the effect on the brain?

Lines 53-54: “The exocyst complex comprises eight subunits, including Sec3, Sec5, Sec6, Sec8, 53 Sec10, Sec15, Exo70 and Exo84, also known as Exoc 1–8” – It is not clear why Sec genes are mentioned. Even if the authors consider it important to leave the mention of these genes, this sentence should still be rewritten so that it is clear that Sec genes are orthologs of mouse Exoc genes. It would also be logical to put the following sentence (Members of the exocyst complex…) before this one.

Lines 69- 70: “Exoc3l2 KO embryos died in utero and showed hemorrhage and hypoplastic hearts. “ -  Written in such a way that the reader should understand that all other organs were normal. It is necessary to refine the text so as not to mislead readers.

Line 175: “(Comscantechno.)” – it is necessary to check the spelling and indicate the country of origin.

 Line 191, 194, 196, 226, 228, 232 and several other lines and Figures along the text: “wild-type (WT) fetus” – the term wild-type (WT) suggests that mice are taken from the wild. It is necessary to write in the Methods which mice were used as control mice and name them accordingly in the text of the Results.

 Lines 187-199:  This text and results are outside the scope of this manuscript and in particular subsection 3.1. Generation of Exoc3l2 KO mice

Figure 1b: in the legend, write what HR and NHEJ and KI (KI allele) are. In addition, it is not clear why the allele delta is shown in the figure if it is not mentioned anywhere else.

Lines 204-205: “Schematic representation of mutant  alleles for Exoc3l2 (b) and Exoc3l3 (c) in mouse.“ – figure 1c does not show a schematic representation of the mutant alleles.

Line 209: “(e) Flowcytometry analysis of Exoc3l2+/GFP embryos at E9.5 with an anti-CD31 antibody.“ – Figure 1e is missing. It is necessary to indicate in the legend what the numbers (79.0, 17.7, 0.24 and 3.03) in Figure 1e mean.

Line 221: “This analysis found that 15% of CD31-positive cells expressed GFP (Fig. 1d).“ – it is necessary to explain to the reader whether this is a lot or a little, and what biological conclusion the authors draw from this fact.

 Line 225:  if the authors write about the Mendelian ratio, then they must demonstrate the chi-square test

Lines 242-243, and 273-274: “Numbers without bracket indicates no dead and hemorrhaging embyro.” – the meaning of this phrase is ambiguous. This sentence needs to be rephrased. Did the authors mean alive embryos without hemorrhaging?

Line 247: ColIV   in the figure is shown as  CollIV

Lines 255-256: “a floxed allele for Exoc3l2 was generated by introducing loxP sequences at both ends of exon 3 (Fig. 3a).” – it is necessary to explain why exactly exon 3 was chosen, to show the result of floxing in Figure 3a, indicating the position of the stop codon. Convince the reader that the presented variant of floxing results in loss of Exoc3l2 expression.

Line 256: “Crossing floxed Exoc3l2 mice with Tie2-Cre mice (expressing Cre in hematopoietic and endothelial lineages” – it is necessary to indicate the designation of the genotypes of crossed mice in terms presented in Figure 3c

 Line 258: “Exoc3l2ΔHEC), which lack Exoc3l2 in hematopoietic and endothelial lineages” – From what is presented in the manuscript, it is not clear why such mice lack Exoc3l2 if they lose only the third exon. Individual exons are often lost during splicing, but this does not always mean loss of function. Explain to the reader why the function is lost in this case.

Line 261: “Exoc3l2ΔHEC embryos showed embryonic lethality around E17.5” – repetition of text in line 260

Line 268, 284: “Hemorrhaging in embryos lacking exocyst complex component 3-like member 2 in hematopoietic and endothelial lineages (denoted Exoc3l2ΔHEC)” – see comment on line 258  

Line 272: “Tie2-Cre:Exoc3l2+/Δ “ - it is necessary to describe or provide a schematic representation of this genotype

Line 275: CollIV  and also in the picture CollIV – make uniform throughout the text (better to use ColIV))

Line 275: it is necessary to write in the legend to figure 3e what the white arrows indicate

 Line 276: “anti-LYVE1” – occurs in the text of the manuscript only here. It is necessary to describe in the Methods and indicate what it was used for

Line 280: the sentence should be reformulated so that references to the figure are consistent, i.e. started from 4a

Line 293: “3.4. Postnatal roles of Exoc3l2 in retinal vascular development” – It is necessary to justify the performance of this part of the work, since it is believed that normally, the complete formation of the vessels of the retina of the eye occurs by the time of birth. Why did the authors expect to see the effect of Exoc3l2 expression deletion in the postnatal period?

Line 297: “were crossed to conditionally delete Exoc3l2” – see comment on line 258  

Lines 324-326: “Some Exoc3l2ΔHEC KO embryos showed ventricular septal defects (Fig. 6b, arrows) and hypoplasia of myocardium (Fig. 6b, c, arrowheads) in the heart at E12.5.” – the word "some" is not allowed. Statistical information is required.

 Lines 327-328: “Exoc3l2ΔHEC KO embryos had morphological brain anomalies in the telencephalon and cer-327 ebellum at E15.5” – it is necessary to describe the found morphological brain anomalies.

Line 344: the telencephalon and cerebellum - different structures of the brain are best shown in different colors. Other brain structures of the experimental animal also do not look the same as in the picture of the control mouse. Why don't the authors compare other brain structures?

 Line 362: after the first sentence, which already talks about abnormal heart development, the word further” is not appropriate in the second sentence.

Line 364: “but not for postnatal retinal angiogenesis.” –  see comment on line 293  

Discussion: Since the authors demonstrated in their work that the lack of Exoc3l2 function in hematopoietic and endothelial lineages is indispensable for normal angiogenesis, it is logical to assume that the absence of the function of this gene should affect not only the development of the cardiovascular system, but also the development and functions of the whole organism and all organs without exceptions. The authors should discuss this.

Author Response

To Reviewer #2

The manuscript presents the results of a study aimed at elucidating the role of Exocyst complex component 3-like 2 on cardiovascular development in mice. The authors used state-of-the-art technology and created knockout mouse models that can serve as experimental models for the Dandy-Walker malformation in Meckel-Gruber syndrome. The manuscript deserves publication, but requires significant revision.

The present version of the title, conclusions, and overall text implies that the effect of the Exoc3l2 deletion extends only to the cardiovascular system and nothing else, but this cannot be the case based on the demonstrated functions of Exoc3l2. The content of the manuscript needs to be reviewed and the emphasis in the presentation of the results, conclusions, and title of the manuscript should be changed so that the reader can understand that the authors consider the cardiovascular system as an example of one of many tissues for demonstrating the function of the Exoc3l2 gene during the development of the organism.

Abstract: The manuscript presents results concerning not only the cardiovascular system, but also the development of brain structures. Why is not a word written in the abstract about the effect on the brain?

We thank for the comment. We have modified the abstract.

Lines 53-54: “The exocyst complex comprises eight subunits, including Sec3, Sec5, Sec6, Sec8, 53 Sec10, Sec15, Exo70 and Exo84, also known as Exoc 1–8” – It is not clear why Sec genes are mentioned. Even if the authors consider it important to leave the mention of these genes, this sentence should still be rewritten so that it is clear that Sec genes are orthologs of mouse Exoc genes. It would also be logical to put the following sentence (Members of the exocyst complex…) before this one.

We thank for the comment. We have modified it as following; The exocyst complex comprises eight subunits, Exoc 1–8.

Lines 69- 70: “Exoc3l2 KO embryos died in utero and showed hemorrhage and hypoplastic hearts. “ -  Written in such a way that the reader should understand that all other organs were normal. It is necessary to refine the text so as not to mislead readers.

We thank for the comment. We have modified it as following; Most of the Exoc3l2 KO embryos died in utero and showed hemorrhage, abnormal heart and brain development.

Line 175: “(Comscantechno.)” – it is necessary to check the spelling and indicate the country of origin.

We apologize for the error. We have corrected it.

 Line 191, 194, 196, 226, 228, 232 and several other lines and Figures along the text: “wild-type (WT) fetus” – the term wild-type (WT) suggests that mice are taken from the wild. It is necessary to write in the Methods which mice were used as control mice and name them accordingly in the text of the Results.

We thank for the comment. We have added the description in Materials and Methods.

 Lines 187-199:  This text and results are outside the scope of this manuscript and in particular subsection 3.1. Generation of Exoc3l2 KO mice 

We thank for the comment. We have modified the subtitle as following; Expression of Exoc3l members and generation of Exoc3l2 KO mice.

Figure 1b: in the legend, write what HR and NHEJ and KI (KI allele) are. In addition, it is not clear why the allele delta is shown in the figure if it is not mentioned anywhere else.

We thank for the comment. We have added description about HR, NHEJ and KI. The allele delta is used in Fig.3 and Fig.5.

Lines 204-205: Schematic representation of mutant  alleles for Exoc3l2 (b) and Exoc3l3 (c) in mouse.“ – figure 1c does not show a schematic representation of the mutant alleles.

We apologize for the error. We have corrected it.

Line 209: “(e) Flowcytometry analysis of Exoc3l2+/GFP embryos at E9.5 with an anti-CD31 antibody.“ – Figure 1e is missing. It is necessary to indicate in the legend what the numbers (79.0, 17.7, 0.24 and 3.03) in Figure 1e mean.

We apologize for the error. We have corrected it.

Line 221: “This analysis found that 15% of CD31-positive cells expressed GFP (Fig. 1d).“ – it is necessary to explain to the reader whether this is a lot or a little, and what biological conclusion the authors draw from this fact.

We thank for the comment. We have added description about the percentage of CD31-positive cells.

 Line 225:  if the authors write about the Mendelian ratio, then they must demonstrate the chi-square test

We thank for the comment. We have added description about chi-square test.

Lines 242-243, and 273-274: “Numbers without bracket indicates no dead and hemorrhaging embyro.” – the meaning of this phrase is ambiguous. This sentence needs to be rephrased. Did the authors mean alive embryos without hemorrhaging?

We thank for the comment. We have added description that the number indicates alive embryos without hemorrhaging in Fig.2 and Fig.3.

Line 247: ColIV   in the figure is shown as  CollIV

We apologize for the error. We have corrected it.

Lines 255-256: “a floxed allele for Exoc3l2 was generated by introducing loxP sequences at both ends of exon 3 (Fig. 3a).” – it is necessary to explain why exactly exon 3 was chosen, to show the result of floxing in Figure 3a, indicating the position of the stop codon. Convince the reader that the presented variant of floxing results in loss of Exoc3l2 expression.

We thank for the comment. We have added a description why exon3 was chosen as following; Exon3 was floxed, because deletion of exon3 leads to a frameshift and production of a truncated protein (209 amino acids) lacking entire Sec6 domain which is conserved among Exoc genes.

Line 256: “Crossing floxed Exoc3l2 mice with Tie2-Cre mice (expressing Cre in hematopoietic and endothelial lineages” – it is necessary to indicate the designation of the genotypes of crossed mice in terms presented in Figure 3c

We thank for the comment. We have added description as following; Crossing Exoc3l2 floxed/floxed mice with Tie2-Cre :: Exoc3l2+/Δ mice (expressing Cre,,,).

 Line 258: “Exoc3l2ΔHEC), which lack Exoc3l2 in hematopoietic and endothelial lineages” – From what is presented in the manuscript, it is not clear why such mice lack Exoc3l2 if they lose only the third exon. Individual exons are often lost during splicing, but this does not always mean loss of function. Explain to the reader why the function is lost in this case.

We thank for the comment. As we explained earlier, we have added a description why exon3 was chosen as following; Exon3 was floxed, because deletion of exon3 leads to a frameshift and production of a truncated protein (209 amino acids) lacking entire Sec6 domain which is conserved among Exoc genes.

Line 261: “Exoc3l2ΔHEC embryos showed embryonic lethality around E17.5” – repetition of text in line 260

We apologize for the error. We have corrected it.

Line 268, 284: “Hemorrhaging in embryos lacking exocyst complex component 3-like member 2 in hematopoietic and endothelial lineages (denoted Exoc3l2ΔHEC)” – see comment on line 258  

We thank for the comment. We have added description as following; Crossing Exoc3l2 floxed/floxed mice with Tie2-Cre :: Exoc3l2+/Δ mice (expressing Cre,,,).

Line 272: Tie2-Cre:Exoc3l2+/Δ “ - it is necessary to describe or provide a schematic representation of this genotype

We thank for the comment. We have added description as following; Exoc3l2 heterozygotes (Exoc3l2+/Δ) expressing Cre under Tie2 promoter.

Line 275: CollIV  and also in the picture CollIV – make uniform throughout the text (better to use ColIV))

We apologize for the error. We have corrected it.

Line 275: it is necessary to write in the legend to figure 3e what the white arrows indicate

 We apologize for the error. We have corrected it.

Line 276: “anti-LYVE1” – occurs in the text of the manuscript only here. It is necessary to describe in the Methods and indicate what it was used for

 We apologize for the error. We have added the description.

Line 280: the sentence should be reformulated so that references to the figure are consistent, i.e.started from 4a

 We apologize for the inconsistency. We have corrected it.

Line 293: “3.4. Postnatal roles of Exoc3l2 in retinal vascular development” – It is necessary to justify the performance of this part of the work, since it is believed that normally, the complete formation of the vessels of the retina of the eye occurs by the time of birth. Why did the authors expect to see the effect of Exoc3l2 expression deletion in the postnatal period?

 We thank for the comment. It is well known that retinal angiogenesis starts from the birth and thus is an excellent model to study angiogenesis during postnatal stage. We have added the description as following; To investigate a role of Exoc3l2 on angiogenesis during postnatal development, we have used an angiogenesis model in retina, because retinal angiogenesis starts from the birth and often used to measure the angiogenic ability of endothelial cells.

Line 297: “were crossed to conditionally delete Exoc3l2” – see comment on line 258  

 We thank for the comment. We have added the description as following; VE-cadherin-CreER :: Exoc3l2+/GFP mice and Exoc3l2 floxed/floxed mice were crossed,,,,

Lines 324-326: “Some Exoc3l2ΔHEC KO embryos showed ventricular septal defects (Fig. 6b, arrows) and hypoplasia of myocardium (Fig. 6b, c, arrowheads) in the heart at E12.5.” – the word "some" is not allowed. Statistical information is required.

We thank for the comment. We have added the description as following; Some Exoc3l2ΔHEC KO embryos showed ventricular septal defects (2 out of 3 examined) (Fig. 6b, arrows) and hypoplasia of myocardium (1 out of 3 examined) (Fig. 6b, c, arrowheads) in the heart at E12.5.

Lines 327-328: “Exoc3l2ΔHEC KO embryos had morphological brain anomalies in the telencephalon and cer-327 ebellum at E15.5” – it is necessary to describe the found morphological brain anomalies.

We thank for the comment. We have added the description

Line 344: the telencephalon and cerebellum - different structures of the brain are best shown in different colors. Other brain structures of the experimental animal also do not look the same as in the picture of the control mouse. Why don't the authors compare other brain structures?

 We thank for the comment. However, we will address the issue in next paper

Line 362: after the first sentence, which already talks about abnormal heart development, the word further” is not appropriate in the second sentence.

We thank for the comment. We have deleted it.

Line 364: “but not for postnatal retinal angiogenesis.” –  see comment on line 293  

We thank for the comment. We have answered earlier.

Discussion: Since the authors demonstrated in their work that the lack of Exoc3l2 function in hematopoietic and endothelial lineages is indispensable for normal angiogenesis, it is logical to assume that the absence of the function of this gene should affect not only the development of the cardiovascular system, but also the development and functions of the whole organism and all organs without exceptions. The authors should discuss this.

We thank for the comment. We have added the description as following; Since Exoc genes are implicated in vesicle trafficking, Exoc3l2 may affect various organ development other than cardiovascular and brain development via paracrine manner.

Reviewer 3 Report

The authors submitted a research article in which they examined the roles of exocyst complex component 3-like 2 (Exoc3l2) during development in mice. They found that Exoc3l1, Exoc3l2, Exoc3l3 and Exoc3l4 are expressed abundantly in endothelial cells and that generation of Exoc3l2 knock-out mice led to disruption of Exoc3l2 resulted in lethal in utero and showed hemorrhage and hypoplastic hearts. These are intriguing results that I found extremaly important for explanation of underlying mechanisms regulating postnatal angiogenesis in
the retina. The aim of the study is clear. The manuscript has a logical structure, well written and conteins well-balanced subsections with findings that cover all aspects of initial hypothesis. The section Discussion along with conclusive part compose off attractive issue for readers. The tables and figures are clear and legible. I congratulate the authors on the study. Although I found the results impressive, I have only one proposal to aothurs. Please, add a brief comment in the section Discussion and 1-2 sentences in the conclusive part to determin the role of the mechanism in discovering tatrgets for further therapy.

Author Response

To Reviewer #3

The authors submitted a research article in which they examined the roles of exocyst complex component 3-like 2 (Exoc3l2) during development in mice. They found that Exoc3l1, Exoc3l2, Exoc3l3 and Exoc3l4 are expressed abundantly in endothelial cells and that generation ofExoc3l2 knock-out mice led to disruption of Exoc3l2 resulted in lethal in utero and showedhemorrhage and hypoplastic hearts. These are intriguing results that I found extremaly important for explanation of underlying mechanisms regulating postnatal angiogenesis in the retina. The aim of the study is clear. The manuscript has a logical structure, well written and conteins well-balanced subsections with findings that cover all aspects of initial hypothesis. The section Discussion along with conclusive part compose off attractive issue for readers. The tables and figures are clear and legible. I congratulate the authors on the study. Although I found the results impressive, I have only one proposal to aothurs. Please, add a brief comment in the section Discussion and 1-2 sentences in the conclusive part to determin the role of the mechanism in discovering tatrgets for further therapy.

We thank for the comment. We have added the description as following; Taken together, we have shown that Exoc3l2 is essential for cardiovascular and brain development. It will be interesting to investigate the molecular mechanism by Exoc3l2 and apply the knowledge to therapy for human diseases.

Round 2

Reviewer 2 Report

the authors finalized the text of the manuscript in accordance with the comments made.